# Bayesian generative models for knowledge transfer in MRI semantic segmentation problems

**Anna Kuzina**                                              A.KUZINA@SKOLTECH.RU

**Evgenii Egorov**                                          EGOROV.EVGENYY@YA.RU

**Evgeny Burnaev**                                          E.BURNAEV@SKOLTECH.RU

*Skolkovo Institute of Science and Technology,*

*Center for Computational and Data-Intensive Science and Engineering, Moscow, Russia*

## Abstract

Automatic segmentation methods based on deep learning have recently demonstrated state-of-the-art performance, outperforming the ordinary methods. Nevertheless, these methods are inapplicable for small datasets, which are very common in medical problems. To this end, we propose a knowledge transfer method between diseases via the Generative Bayesian Prior network. Our approach is compared to a pre-train approach and random initialization and obtains the best results in terms of Dice Similarity Coefficient metric for the small subsets of the Brain Tumor Segmentation 2018 database (BRATS2018).

**Keywords:** Brain Tumor Segmentation, Brain lesion segmentation, Transfer Learning, Bayesian Neural Networks, Variational Autoencoder, 3D CNN, Variational Inference

## 1. Introduction

Semantic segmentation of MRI scans is an essential but highly challenging task. State-of-the-art methods for semantic segmentation imply the use of DNN, which usually have millions of tuning parameters, hence demanding a large amount of labelled training samples.

Manual labelling of the MRI with tumor is time consuming and expensive. Hence, in most cases only tiny datasets are available for training. To improve the model performance, we can exploit knowledge from existing labelled datasets. Nevertheless, these images may be pretty different in terms of diseases, modality, protocols and preprocessing methods, which leads to extra difficulties.

In this work, we address the problem of knowledge transfer between medical datasets when source dataset potentially contains relevant information for the given problem (e.g. it depicts scans of the same organ), but still comes from the different domain, complicating the work of the conventional transfer learning techniques. The main contributions of the paper are the following:

- We highlight that the fine-tuning (start training with the pre-trained model) is not applicable for medical images

- We propose using Bayesian approach with implicit generative prior in the space of the convolution filters instead of simple fine-tuning

- Results are validated, using BRATS18 (Menze et al., 2015; Bakas et al., 2017) and Multiple Sclerosis Human Brain MR Imaging (MS) (CoBrain analytics) datasets

## 2. Method

Transfer learning is a set of techniques from machine learning, used to store knowledge from source dataset and apply it to the related target dataset. During our experiments with MRI semantic segmentation, we have noticed that kernels from different segmentation networks share a similar structure, when appropriately trained, in contrast to noisy kernels from models trained on small datasets. Therefore, prior distribution, which restricts kernels to be more structured, presumably, should improve segmentation quality on modest training sets. We propose to apply Deep Weight Prior (Atanov et al., 2018) to enforce precisely this property. Deep Weight Prior is an expressive prior distribution, which helps to incorporate information about the structure of previously learned convolutional filters during the training of a new model. We will consider implicit prior distribution in the form of Variational Autoencoder (VAE) (Kingma and Welling, 2014) with encoder $r_{\psi^{(i)}}(x|w)$ and decoder $p_{\phi^{(i)}}(w|z)$, modeled by neural networks.

The approach has the following steps:

1. Given source dataset $D_1$, train the DNN model and collect dataset of the convolutions filters during training.

2. Train VAE on the collected dataset of the of the convolutions filters.

3. Perform the variational inference for target dataset $D_2$ using VAE as the prior over the model filters.

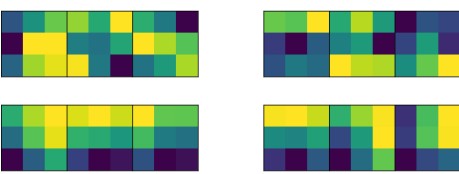

Left: Kernels from U-Net, trained on the source MS dataset. Right: Samples from the trained DWP

U-Net (Ronneberger et al., 2015) was chosen due to its popularity and experimentally proven efficiency for MRI semantic segmentation tasks (Deniz et al., 2018; Livne et al., 2019; Guerrero et al., 2018; Milletari et al., 2016). The chosen architecture has 726480 parameters. We denote by $w^{(i)}$, $i = 1, ..., L$ kernels for the $i$th convolutional layer and by $w = (w^{(1)}, \ldots, w^{(L)})$ vector of all the model parameters. If kernel filters at a layer $i$ are of size $3 \times 3 \times 3$, with $C_{inp}^{(i)}$ input channels and $C_{out}^{(i)}$ output channels, then the weight matrix has dimensions of $C_{inp}^{(i)} \times C_{out}^{(i)} \times 3 \times 3 \times 3$. We assume that both variational approximation $q_\theta(w)$ and prior distribution $p(w)$ are factorized over layers, input and output channels. with encoder $r_{\psi^{(i)}}(x|w)$ and decoder $p_{\phi^{(i)}}(w|z)$, modeled by neural networks. Finally, we need to optimize over $\theta, \psi$ to learn the model with the DWP-prior using the following loss:

$$\max_{\theta,\psi} \mathcal{L}^{\text{approx}} = \max_{\theta,\psi} \mathcal{L}_\mathcal{D} + \sum_{p,k,i} \left[ -\log q_{\theta_{ipk}}(\widehat{w}_{p,k}^{(i)})) - \log r_{\psi^{(i)}}(\widehat{z}|\widehat{w}_{p,k}^{(i)}) + \log p(\widehat{z}) + \log p_{\phi^{(i)}}(\widehat{w}_{p,k}^{(i)}|\widehat{z}) \right],$$

where $\mathcal{L}_\mathcal{D}$ is the likelihood of the selected model. The methodology is discussed in more details in Kuzina et al. (2019).

## 3. Experiments and Results

The experiments aim at comparing the proposed method (UNet-DWP) with the conventional transfer learning approaches: training the whole model on the small target dataset with the weights pretrained on the source dataset (UNet-PR) or freezing layers in the middle of the network (UNet-PRf) while fine-tuning only the first and the last block of the model to reduce overfitting on a small dataset. As a baseline, we also consider random initialization (UNet-RI), where the model is trained only on the small target dataset. To compare the proposed methods, we use MS dataset (CoBrain analytics) as a source and small subsets of BRATS18 dataset (Menze et al., 2015; Bakas et al., 2017) as targets. Both datasets consider the MRI scans of the brain, however, with different diseases. The purpose of this setup is to show the ability of the method to generalize between diseases.

Models performance was compared on the whole tumour segmentation on subsets of BRATS18 volumes, containing 5, 10, 15 or 20 randomly selected images with the fixed test sample size of 50 images. To train U-Net in the non-Bayesian setting, we use a combination of binary cross-entropy and Dice losses. Each model was estimated at three different random train/test splits. Table 1 summarizes the obtained results [1].

### 3.1. Results

We can see that the models trained with DWP noticeably outperform both randomly initialized and pre-trained U-Net for all the training sizes. We observe higher variability in prediction accuracy for the problems with smaller sample sizes, which shrinks as training dataset grows, and the superiority of UNet-WDP becomes clearer. It is also worth mentioning that the pre-trained model, where part of the weights was frozen, fails. We believe that this means that information from other diseases is not relevant for the new task by default, and without fine-tuning of the whole network, we are not able to achieve consistent results.

| Train size | UNet-DWP (ours) | UNet-PR | UNet-PRf | UNet-RI |
|---|---|---|---|---|
| 5 | **0.52** (0.05) | 0.49 (0.02) | 0.45 (0.03) | 0.50 (0.02) |
| 10 | **0.58** (0.05) | 0.52 (0.01) | 0.47 (0.03) | 0.53 (0.01) |
| 15 | **0.60** (0.02) | 0.56 (0.02) | 0.50 (0.02) | 0.58 (0.02) |
| 20 | **0.63** (0.01) | 0.58 (0.01) | 0.53 (0.02) | 0.60 (0.01) |

Table 1: Intersection over Union metrics for the experiments with small available target dataset.

It is worth mentioning, that transfer learning model on average performs even worse than the model without any prior knowledge about the data. This result is quite surprising, but it can be explained by strong disease specificity of the data. Datasets differ not only in the shapes of the target segmentation (plaques of multiple sclerosis are much smaller and difficult to notice that brain tumour) but also in resolution, contrast and preprocessing method. As a result, after corresponding initialization, fine-tuning may converge to a worse solution.

---

1. An implementation of the methods can be found at https://github.com/AKuzina/DWP

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
