# OpenReview forum: "Bayesian Generative Models for Knowledge Transfer in MRI Semantic Segmentation Problems"
_MIDL.io/2020/Conference — MIDL 2020_

### Official Review · AnonReviewer2 · 2020-03-09
**Transferring knowledge using Deep Weight Prior**

**Rating:** 3
**Confidence:** 3

**Review:**

The paper proposes to apply Deep Weight Prior to the problem of transfer learning in medical imaging. The authors learn a U-Net on MS lesion segmentation and evaluate transferability to the BRATS2018 dataset. The use of DWP is well motivated and the results indicate improved performance over regular transfer learning.

Following the results of the paper, I believe the use of DWP can improve settings in medical image analysis with only limited available training data but availability of related datasets. However, the authors should improve the explanation of DWP and introduce the used variables. For example, I assume that $p, k$ in the equation on page 2 refers to the input and output  channel dimensions of the convolutional kernels. It would be interesting to report Dice scores which are more commonly used for the BRATS dataset. It would be beneficial for the authors to release code or add training details to the appendix as the results seem to be irreproducible in its current form. Lastly, a longer study should test different freezing regimes for transfer learning, as freezing the middle seems like a rather arbitrary choice.

Minor:
- page 2, 1. dataset instead of dataest
- page 2, the figure and the enumeration could use a little margin between them

---

### Official Review · AnonReviewer3 · 2020-03-13
**A nice application of DeepWeightPrior**

**Rating:** 4
**Confidence:** 5

**Review:**

The authors use deep weight prior to learn an implicit prior distribution over the weights to facilitate transfer learning.
This allows the model to mitigate overfitting on the target task when limited labeled data is available.

To evaluate, an MS dataset was selected as the source and small subsets ofBRATS18 dataset were selected as target. The evaluation was performed on a fixed number of target images but when having access to varying number of labeled data from the target domain.

---

### Official Review · AnonReviewer1 · 2020-03-14
**Learning prior distribution of CNN weights as a pretraining**

**Rating:** 3
**Confidence:** 4

**Review:**

The authors present a method to pre-train deep neural architectures for the purpose of medical image segmentation in the situation of small training datasets. This method is based on learning prior distribution of the CNN weights based on a generative model referred to as deep weight prior (DWP) proposed by Atanov et al. The authors propose to learn the kernel distribution from a source dataset consisting of MRI of multiple sclerosis (MS) patients and apply it to the task of segmenting brain tumors from the BRATS18 MRI dataset considered as the target domain. UNet is used as the backbone architecture.
The proposed method is compared to three baseline methods, namely a model directly trained on the low sample target data (BRATS 18) based on standard random initialization (UNet-RI), a model whose weights are pre-trained on the MS Datasets (UNet-PR), and the UNet-PR model fine-tuned on the BRATS18 dataset (UNet-Prf).
Results based on the intersection over union metric indicate that the model performs better that UNet-PR and UNet-PRf but comparably with UNet-RI.
I have one major concern regarding the validity of the hypothesis grounding the DWP method proposed by Atanov et al. The authors indeed assume that the source and target kernels (networks weights) are drawn from the same distribution, so that the source kernel distribution that is learned can serve to perform Bayesian inference on the target data.  I am not sure that this assumption holds for the source (MS) and target data (BRATS). The diagnostic tasks are indeed very different, so that, I guess, the kernels are likely to differ. I am not sure that the DWP is the best adapted for this specific transfer learning task. This may explain, why, as suggested by the authors, UNet-DWP does not perform much better than the random initialization (UNet-RI).
Description of the DWP method as well as the source (MS) and target (BRATS18) datasets should be detailed, extracting some more details from the recently published paper in Frontiers in Neuroscience.

---

### Official Review · AnonReviewer4 · 2020-03-14
**The paper proposes a way of training deep segmentation networks on small medical datasets by learning a prior distribution on convolution kernels**

**Rating:** 4
**Confidence:** 4

**Review:**

The idea of learning prior to distribution on convolution kernels is methodologically sound and appealing. This new way of transfer learning could potentially be more effective than fine-tuning and L2 regularization (which basically is a zero-mean Gaussian prior). The preliminary results are reasonable.

Now the authors should think about how to further extend and validate this work in the following two aspects:

1. how can the generative power of VAE be used in the segmentation model? Can you learn a family of DNNs to improve segmentation or quantify uncertainty?

2. how does the prior compare to other standard regularization approaches?

---

### Meta-Review · Area_Chair1 · 2020-04-06
**MetaReview of Paper256 by AreaChair1**

**Rating:** 4

**Metareview:**

All the reviewers recommended acceptance of this work. I agree with them in that it is an interesting work and should be accepted as a short paper in MIDL 2020.

The reviewers have raised a few points that would be interesting discussing in the final camera ready version. Please, when submitting the final manuscript, try to to address these points.

**Paper Type:**

methodological development

---

### Decision · Program_Chairs · 2020-04-11

Accept